# Emerging Silk Material Trends: Repurposing, Phase Separation and Solution-Based Designs

**DOI:** 10.3390/ma14051160

**Published:** 2021-03-01

**Authors:** F. Philipp Seib

**Affiliations:** Strathclyde Institute of Pharmacy and Biomedical Sciences, University of Strathclyde, Glasgow G4 0RE, UK; philipp.seib@strath.ac.uk

**Keywords:** silk fibroin, liquid–liquid phase separation, *Bombyx mori*, medical silks

## Abstract

Silk continues to amaze. This review unravels the most recent progress in silk science, spanning from fundamental insights to medical silks. Key advances in silk flow are examined, with specific reference to the role of metal ions in switching silk from a storage to a spinning state. Orthogonal thermoplastic silk molding is described, as is the transfer of silk flow principles for the triggering of flow-induced crystallization in other non-silk polymers. Other exciting new developments include silk-inspired liquid–liquid phase separation for non-canonical fiber formation and the creation of “silk organelles” in live cells. This review closes by examining the role of silk fabrics in fashioning facemasks in response to the SARS-CoV-2 pandemic.

## 1. Introduction

More than 1.5 million different medical devices are in use today, and yet the palette of materials approved for human use is extremely small [1]. This places limits on medical progress and has created a strong demand for an orthogonal strategy for the design and discovery of novel materials, including new silks.

“Everyday” silk from the silkworm *Bombyx mori* is well-known for its use in clothing. However, silk offers solutions to many biological challenges (e.g., housing, protection and on-demand assembly) because it has arisen in nature at least 23 times in independent convergent evolutionary events in a range silkworms, spiders and other organisms, and its ubiquity and widespread use are clear testaments to its biological success [2]. Clinically, silk offers solutions to unmet healthcare challenges and contributes towards a healthy nation. Silk is a high-performance biomaterial that is already clinically approved due to its renowned biocompatibility, low immunogenicity and tunable biodegradation (minutes to years) [3]. Silk degrades into benign products that do not accumulate in the body or the environment [4]. Its unique physical properties (e.g., toughness) support the medical use of silk as a suture material and surgical mesh for load-bearing applications (Sofregen Inc.) [3,5]. Silk’s robust safety record in humans and in ongoing clinical trials makes it a highly attractive material for state-of-the-art medical applications [3]. The spun silk fiber can either be used directly or reverse engineered into liquid silk for processing into stimulus-responsive nanomedicines [6], stabilizers for payloads (drugs, proteins and diagnostics) [7], medical sensors [8], hydrogels for tissue engineering [9] and vehicles for drug and cell delivery [6,10,11].

While the biomedical use of silk has spanned several millennia, silk did fall out of fashion with the advent of synthetic manmade fibers, in the erroneous belief that we are able to “beat” nature. However, over the past 20 years, interest has been renewed both in our fundamental understanding of silk and in its biomedical applications [12,13]. Today, more than 13,000 publications on this subject are at our fingertips, and the number continues to grow (PubMed accessed January 2021). At the same time, technological innovations [14] open up new silk processing strategies, uses and applications. A number of timely reviews have covered the biomedical use of silk [3,12], recombinant silks [15] and their applications (e.g., drug delivery [16], tissue engineering [17], additive manufacturing [18] and disease models [19]). For example, an excellent review of silk nanoparticles is included in this special issue on silk-based biomaterials [20]. Silk is also making a marked ingress into the cosmetics and personal care products industry. Here, recombinant spider silk-inspired proteins are often used to ensure vegan product certification (reviewed in Reference [21]).

In this review, I have selected key primary research papers published in 2020 (references [22,23,24,25,26,27,28,29,30,31]) that I believe will change the silk material space (Figure 1). Typically, technological innovations have catalyzed the fields that ultimately delivered the most impact. For example, the desire to “unspin” the silk fiber was first reported in the 1930s, with the use of the now common lithium bromide approach [3]. This approach was subsequently refined [14] and ultimately led, in the 2000s, to a prolific number of studies and patents applying this technology to a wide spectrum of medical applications [3,32]. This knowledge has subsequently been used to found several companies that are developing products that are now finding their way into the marketplace [3]. For example, in 2019, Sofregen Inc. (Medford, MA, USA) received the first Food and Drug Administration clinical approval for their silk hydrogel for vocal fold augmentation (Silk Voice^®^, Medford, MA, USA). This has been a key milestone because Silk Voice^®^ is manufactured from processed liquid silk. Silk Voice^®^ is therefore a first-in-class product that uses “unspun” silk. Based on this success, more products are expected to enter clinical trials and ultimately progress into routine clinic use. To date, a majority of studies have used “everyday” silk from the domesticated silkworm *Bombyx mori*, because the established silk-farming (i.e., sericulture) industry provides a dependable supply chain. However, recombinant silks are particularly valuable for the rapid creation of bespoke, high-fidelity materials, while the first example of artificially created living silk materials [31] will add a completely new dimension. All naturally occurring silks have clear differences, but they also share common design features (e.g., storage stability, self-assembly on demand and robust mechanics). Therefore, recognizing the principles that apply to silks more broadly is particularly valuable because it enables rapid adoption across the material community that is working with different silks. Ongoing efforts to create genetically engineered silkworms will continue to expand the capabilities of silk (e.g., Reference [33]) and to blur the lines between the different silk types (e.g., *Bombyx mori* versus spider silk). These sericulture efforts will ultimately provide a simple but scalable manufacturing route for high-value products at a very large scale [34]. The first products derived from recombinant silks in vitro are now entering the consumer market [21]. Importantly, the pharmaceutical industry is accustomed to working with recombinant proteins and will therefore be open to the rapid adoption of this manufacturing approach. In this review, I provide timely examples, using both everyday and recombinant silks (Figure 1). I use the term silk to refer to *B. mori* silk fibroin, unless stated otherwise.

## 2. Silk Fundamentals: Flow and Liquid–Liquid Phase Separation

Nature has evolved elegant and effective solutions to create silks that are processed and spun by silkworms, spiders and other organisms, on demand, with minimal energy expenditure [35]. Silk is a renewable protein block copolymer. Many properties of silk arise from the unique amphiphilic structure of its heavy chain. For example, *B. mori* silk includes 11 hydrophilic blocks and 12 hydrophobic domains. The hydrophobic blocks account for 94% of the chain sequence and contain highly repetitive glycine-X repeats, where X is alanine (A) (65%), serine (S) (23%) or tyrosine (Y) (9%) [36]. During the fifth and final instar, the silkworms synthesize large amounts of silk over several days. This silk is stored in liquid gel-like form in the silk gland, until the silkworm is ready to spin its cocoon. The silkworm spins this liquid silk “dope” into a fiber that is then three-dimensionally “printed” over the course of a day into a cocoon. This “printing” is conducted by using aqueous conditions, at ambient pressure and temperature, with no fouling of the spinneret. This processing is in stark contrast with the situation with synthetic fibers. For example, nylon is typically melt spun at an industrial scale, followed by drawing and annealing to improve the fiber characteristics [37]. For silk, the remarkable fiber properties are a product of the protein sequence and the carefully orchestrated assembly of the hierarchal structure of the fiber. To date, the impact of shear [38], pultrusion [7], pH [39,40,41] and extensional flow [42] have contributed to our understanding of silk processing and assembly from a gel-based system to a solid (i.e., protein precipitation). Metal ions are also known to impact fiber characteristics [43], and metal ions change both along the silk gland and during the spinning stage. Therefore, ions are also poised to play a role in experimental silk processing. For example, water removal and preparation of a silk from its stable storage conformation to a form ready for spinning is regulated by several factors, including metal ions. Metal ions ultimately contribute to the beta sheet rich silk fiber [40,44]. However, the flow properties of silk in response to metal ions have been poorly defined.

Flow is the foundation of silk fiber formation; therefore, these types of studies are important. Seminal work by Chris Holland and co-workers, in 2020, now shows the underlying fundamental principles of the effect of metal ions on silk flow [26,29]. Precautions were taken to reduce artefacts. The authors used fastidious isolation of liquid silk directly from the silk gland and ensured handling with minimal mechanical manipulation, so the silk stock faithfully reflected the primary and higher order silk structure. This work is reported across three publications [26,29,45], with the first one appearing in 2018 [45] and detailing a correlation between silk viscosity and calcium and potassium ion concentrations; these are the most abundant ions in the silk gland [45]. These reports provided the first explanation for the variable silk viscosities described previously [46,47]. The 2018 study [45] showed that, before and during cocoon construction, the silk stock changes viscosities in vivo. Therefore, the variability reported in early studies was a reflection of the exquisite responsiveness of silk to in vivo processing rather than to a random sample variability. During cocoon spinning, the middle posterior gland showed a slight increase in calcium ions and a simultaneous dramatic increase in potassium ions from 7 to 50 cations per silk chain. This increase in potassium ions was accompanied by a dramatic drop in viscosity. These findings led to the formulation of the hypothesis that calcium ions would form salt bridges between a total of 77 acidic amino acids present in silk, while potassium ions would compete and lead to a reduced viscosity [45]. The most recent studies using wet lab experimentation [26] and theoretical modeling of “sticky reptation” support this hypothesis [29]. The modeling showed that calcium ions formed salt bridges that restricted chain motion and diffusion, thereby ultimately impeding relaxation due to flow stress and cumulating in increased viscosity. In this model, the silk chain showed approximately 10 entanglements and an equivalent number of temporary sticky crosslinks (“stickers”) via calcium ion-mediated salt bridges. However, most importantly, the sticker lifetime was impacted by the ionic concentration. At higher potassium ion concentrations, the sticker lifetime was reduced; this observation was consistent with the experimental observations of improved flown [29]. Overall, the modeled dataset from experimental measurements was further substantiated.

The third publication details a causative study showing how externally added metal ions impact silk flow properties [26]. Lithium chloride and lithium bromide salts, at the high concentrations often used to solubilize silk during regeneration, substantially reduced molecular interactions, thereby preventing the formation of large aggregates but enabling alignment. By contrast, the addition of calcium ions not only increased viscosity but also impeded shear alignment and self-assembly. Therefore, calcium ions act as a switch that activates the silk storage phenotype, whereas potassium primes silk for alignment and self-assembly. Overall, the knowledge created by these fundamental studies [26,29,45] is likely to impact the engineered silk spinning processes, our understanding of hierarchical biomaterials in general and potentially the design of biomimetic polymers with environmentally controlled functions.

Many studies have set out to emulate silk spinning using various silk stocks or spinning rigs [48]. These studies have benefited from our understanding of how silk spinning in nature is carefully orchestrated to yield a solid high-tech fiber. Molecular biologists might regard this flow-induced transition of silk from a liquid state to a solid fiber as denaturation, while a polymer scientist would be more inclined to label this process as flow-induced crystallization. Although silk spinning is particularly important for processing recombinantly expressed silks, translating natural silk-spinning principles to synthetic polymers is also likely to have a significant impact, due to the current volume and demand for these polymer materials. This area of study will benefit from biomimicry, to support more sustainable growth.

The seminal work published in 2020, by Oleksandr Mykhaylyk and co-workers [22], has now made a significant step into uncharted territory by applying flow-induced crystallization of poly(ethylene) oxide. This work borrows key principles from silk spinning, particularly the phase transition from liquid to solid. For this process, silk does not require the heat transfer or chemical crosslinking typically required for synthetic polymers. So, how does nature overcome this requirement? The answer is aquamelts: flow-induced aggregation of silk from solution into a solid by displacing the hydration layer of energetically bound water that surrounds silk [35]. This flow-triggered solidification enables animals to synthesize and store a liquid silk stock that, on demand, is spun into a solid functional construct (e.g., webs, cocoons, etc.). Priming silk with ions and pH speeds up the nucleation step and contributes to fiber formation by flow at a low energy expense. 

Unlike thermoplastics spinning, natural silk spinning is, by orders of magnitude, a more energetically efficient process and only needs flow at ambient conditions to induce crystal nuclei [35]. Mykhaylyk and co-workers have exploited this process to demonstrate, for the first time, how a metastable poly(ethylene) oxide solution can be converted into a crystalline solid with flow under ambient conditions in the absence of a chemical reaction, removal of heat or evaporation of solvent [22]. This phase transition requires the flow to exceed the required energy threshold that disrupts the protective hydration shell around poly(ethylene) oxide. The flow orients and stretches the polymer along the flow direction, breaking hydrogen bonds between the water and the polymer and rupturing the hydration shell, thereby exposing stretched desolvated poly(ethylene) oxide segments. These exposed segments of similar configuration then self-assemble to form intermolecular interactions that crystallize. While more experimental verification is needed, this mechanism might be broadly applicable to other polymers, including biopolymers (see below). One key requirement is that the polymer also undergoes a specific interaction with the continuous phase, and this depends on the polymer conformation. Overall, flow-induced crystallization improves energy efficiency and demonstrates the orthogonal application of silk-spinning fundamentals to synthetic polymers.

In the silk gland, liquid–liquid phase separation is a hallmark of silk storage [49], while the application of energy via flow is a key trigger for the phase transition to a solid [50]. Details are unravelling regarding silk fiber assembly via liquid–liquid phase separation, nanofibrillation and multiple chemical and physical triggers in different spidroin functional domains [51]. However, a solution-solid transition due to flow is not unique to silk but is applicable to other proteins, some often implicated in disease. Experimental studies by Tuomas Knowles and co-workers, published in 2020, now show that exposing phase-separated protein-loaded droplets to a critical fluid shear of approximately 0.5 Pa generates protein fibers [30]. The required shear stress is relevant for biological systems and highlights the potential danger of irreversible phase transitions in disease. The selected proteins and peptides, with no function associated with fiber formation, readily underwent the liquid-to-solid-fiber transition. Regenerated *B. mori* silk was included as a positive control in their studies. As with silk, all the other fibers were stable for days under the same experimental conditions. One requirement for successful fiber formation was the ability to form highly aligned nanofibrils. These nanofibrils are predicted to be stabilized by backbone–backbone hydrogen bonding that drives the transition from solution to solid. To test this hypothesis, a proline–arginine repeat peptide (Mw 7225 g/mol) was generated and triggered to undergo a liquid–liquid phase transition. However, this proline–arginine repeat peptide was not able to form backbone–backbone hydrogen bonding, so it failed to transition from solution to a fiber. This study demonstrates that flow alignment of proteins also enables their condensation and phase transition [30]. Therefore, the principle of flow-induced crystallization appears to be more broadly applicable than first anticipated. This is important because it opens up new processing technologies for polymers beyond silk.

One mechanism for compartmentalization that is used by both eukaryotic and bacterial cells alike is liquid–liquid phase separation. Phase-separation phenomena enable living cells to control essential biochemical functions (e.g., RNA processing, embryogenesis, etc.) in the absence of a classical membrane [52]. Advances in imaging technologies are revealing an increasing number of bacterial organelles (e.g., carbonosomes, magnetosomes, lipid bodies, etc.) (reviewed in Reference [53]). For example, in rapidly dividing *Escherichia coli* cells, the RNA polymerase itself is subject to liquid–liquid phase separation [54]. Work by Xiao-Xia Xia and co-workers, published in 2020 [31], applied liquid–liquid phase separation to *E. coli* to establish an intracellular liquid silk compartment that could be exploited for synthetic biology [53]. The *E. coli* cells were first genetically modified to express a recombinant form of major ampullate spidroin 1 of the spider *Nephila clavipes* [31]. The recombinant form was expressed both with and without N-terminal fused green fluorescent protein (GFP) via a flexible linker, to permit tracking in live cells. Unlike the GFP control, the silk construct formed compartmental condensates that were localized to the pole regions of live cells. These liquid silk compartments were established under a broad range of conditions and were independent of GFP. Probing with thioflavin T indicated some beta sheet formation within the silk condensates. These silk condensates were liquid, and in vitro studies strongly supported the hypothesis that these condensates were formed by liquid–liquid phase separation. The newly synthesized silk was concentrated in the vicinity of ribosomes that enabled the formation of hydrophobic interchain assemblies, most likely between poly-alanine stretches, resulting in short stabilizing beta sheets. Therefore, the inherent silk property to assemble into liquid–liquid phase separated structures could also be imported into living cells. This study also demonstrated that other unstructured proteins, such as resilin-like protein and major ampullate spidroin 2, were able to create condensates, opening up the experimental space even further. 

The ability to functionalize the silk compartments was proven with the model cargo red fluorescent protein fused to silk and co-expressed in *E. coli* [31]. The protein localized to the common silk compartment. Next, to provide the silk compartment with catalytic activity, metallothionein was fused with silk and expressed. The live cells expressing metallothionein were then exposed to sodium selenite, and they reduced this to form-functional 20 nm metal nanoparticles that were localized to the common silk compartment. Additional experiments to equip the silk compartment with multiple de novo enzymes were attempted, but this approach requires further optimization. Exploiting silk for the de novo creation of a functional compartment within cells is exciting, especially since it appears to have minimal impact on normal cell physiology. This approach [31] now opens the door not only to the creation of novel functional materials in living cells but also to the creation of new intracellular organelles for synthetic biology.

## 3. Silk Processing

Silk fibers spun in vivo and in vitro continue to find new medical applications. Examples include in vitro generated multi-functionalized sutures [55] and custom in vivo spun fibers that are incorporated into resins [56]. A key tool that has made these new features possible is our ability to fully reverse-engineer the spun fiber [14,48]. First, the silk cocoon is degummed and the extracted fibers are dried. Next, the silk fibers are dissolved in lithium bromide, to break the hydrogen-bonding network in the β-sheet nanocrystallites; this step ultimately dismantles the higher-order silk structures. The solubilized silk is then dialyzed against water, to generate the now-popular regenerated silk solution [14]. A number of variations exist for this classical protocol, but all result in the production of an aqueous silk stock. Processing advances over the past 15 years have expanded our toolbox to now include the following features, among others:Tuning of the aqueous silk secondary structure via water vapour annealing, thereby eliminating the need for solvents (e.g., methanol) [57];Production of off-the-shelf lyophilized silk powders for instant reconstitution to reduce the lead time for material formation and substantially extend the silk shelf-life [58];Generation of elastomeric silks by chemical crosslinking of tyrosine generating, optically clear polymeric networks [59];Payload stabilization (e.g., drugs, enzymes and diagnostic samples) that expands the use of silk to drug delivery formulations [60,61];Formation of silk bioinks that broaden the available material palette [18].

The hallmark of this toolbox is the use of regenerated silk that is processed into novel formats (e.g., films, sponges, gels, particles, bioinks, etc.). Phase separation is a common protocol used to create new silk formats, and a broad spectrum of processing tools exists for this purpose (reviewed in Reference [14]). An interesting orthogonal approach to create phase-separated hierarchical structures is the combination of top-down manufacturing and directed silk self-assembly [62]. Here, a silk solution was first chemically crosslinked, followed by application of mechanical tension and then controlled water removal, to align the construct and induce the formation of beta sheets. While this example demonstrates our ability to orchestrate the material properties from the nano- to macro-scales, multiple steps were required to create these materials [62]. Typically, the generation of different material formats requires downstream processing; for example, solvent addition and removal to yield solution-stable silk formats. The resulting materials are useful, but these processing steps are costly and impede commercial translation. Therefore, efforts to simplify silk processing are important to ultimately support product translation.

In this regard, an innovation reported in 2020, by David Kaplan and co-workers, is the use of lyophilized silk powders for thermoplastic molding [24]. Silk fibers are processed into a reconstituted silk solution, which is then lyophilized and milled to generate silk powders for secondary processing. This silk powder is amorphous and contains 5% (w/w) residual water (a significantly higher amount than in degummed native silk fibers). This water serves as a thermal plasticizer, while the absence of a high beta sheet content permits molecular re-arrangements. Thermal analysis showed a water-associated glass transition temperature at 65 °C and a stable glass transition temperature at 178 °C. The high residual water content and amorphous silk state of the starting powder is important because attempts to thermally melt beta-sheet-rich silk typically results in sample burning [24]. An exception has been reported for small sample sizes (<1 mm^2^), using fast scanning chip calorimetry at 2000 k/s [63].

Previous work has shown that silk films could be welded [64] and nanoimprinted, using heat and pressure, by exceeding the water-associated glass transition temperature and triggering the transition from a glassy state to a liquid-like state [65]. At that time, no attempts were made to create bulk materials using this process. However, the thermoplastic molding study conducted in 2020 exposed silk to 632 MPa and different heat regimes and successfully created silk bulk materials that transitioned visibly from white to transparent to pale yellow [24]. At 125 °C, the silk powder particles were closely compacted, whereas they were fused at 145 °C. At these high temperatures, the increased chain mobility and water removal allowed silk to adopt beta sheet–rich structures. These silk bulk materials had excellent mechanical properties (e.g., specific strength 109 MPa g^−1^ cm^−3^). Intriguingly, a catalytic model enzyme included in the silk mix retained its activity even after exposure to the 145 °C pressing condition. 

The pressed silk bulk materials could be easily machined or, through the use of molds, directly formed in situ [24]. This process was demonstrated with the formation of screws intended for orthopedic applications. In vivo studies showed that these silk screws were biocompatible, despite their slow degradation. A cost analysis showed that this novel thermoplastic molding of silk was at least 100 times more cost effective than traditional methods, while also eliminating solvent waste and minimizing occupational solvent hazards. Work conducted around the same time by Tuan and co-workers used milled raw sericin-containing silk and hot pressing (150–180 °C 31 MPa) to create monolithic silk resins [66]. These authors have since applied their technology to regenerated silk powders [67]. Overall, thermoplastic molding of silk adds a novel and useful processing strategy to the current manufacturing toolbox.

## 4. Biomedical Applications 

The popularity of silk continues to grow, impacting virtually all areas of biomedical research. For example, the precision cutting [68] or patterning of silks has already provided the ability to influence applications, ranging from stem cell biology to the design of rewritable optical storage devices [69]. However, in light of the ongoing Covid-19 pandemic, I have picked one example where silk is having a particular impact. At the start of the Covid-19 pandemic, the shortage of personal protective equipment spurred the demand for homemade face masks, to reduce the transmission of severe acute respiratory syndrome coronavirus 2 (SARS-CoV-2) via respiratory droplets. Supratik Guha and co-workers examined the potential of common household fabrics, including silk, for their potential use in face mask construction [27]. To measure the filtration efficiencies, this research group used a custom-built system which included a sodium chloride–particle generator to emulate respiratory droplets. The initial publication reported that multiple layers of fabrics improved filtration efficiency, when compared to a single layer [27]. Furthermore, cotton/silk hybrid fabrics were particularly well suited as mask materials. The work attracted global attention. The authors have since published a correction [70], expanded the dataset and responded [28] to letters to the editor [25,71,72].

Subsequent measurements by other researchers from three different laboratories have demonstrated that natural silk has the lowest impedance, compared to cotton or cotton/polyester mixtures with different thread counts per inch (tpi) [25]. For mask performance, impedance is important because it relates directly to the comfort and breathability of the material. Impedance also impacts the degree of protection offered by the mask. As impedance increases, the unfiltered tributary airflow through leaks in the mask also increases, thereby reducing the overall degree of protection. Meticulous measurements indicated that four layers of silk had a 40 to 50% filtration efficiency, which was substantially higher than four layers of 400 tpi cotton (and these data now correct the scientific literature). While cotton failed the NIOSH 42 CFR 84 standard for breathability, silk passed easily. Silk has the added advantage that it minimizes skin irritation. Nonetheless, both N95 and surgical masks outperformed silk or any other materials tested and therefore remain the preferred choice.

## 5. Conclusion and Outlook

Fashioning facemasks from silk during a global pandemic is a clear testament that the silk fiber continues to support human health in its native form, despite our ability to unspin the silk thread. While the sole focus of this review has been to provide examples of recent advances in fundamental silk science, the application of silk science findings to synthetic polymers, the orthogonal use of silk in living cells and the novel silk processing now available all confirm that the silkworm remains an invaluable asset that goes well beyond its ability to spin the silk thread. Silkworms have evolved to produce large amounts of protein and are thus ideally placed to be genetically re-programmed, allowing *B. mori* silkworms to also serve as biofactories of tailor-made high-value proteins [34,73]. For example, the use of the *B. mori* genetic toolbox made possible the expression the SARS-CoV-2 spike protein in silkworms [23], and the use of silkworms to generate virus-like particles is expected to contribute to the SARS-CoV-2 vaccine pipeline.

## Figures and Tables

**Figure 1 materials-14-01160-f001:**
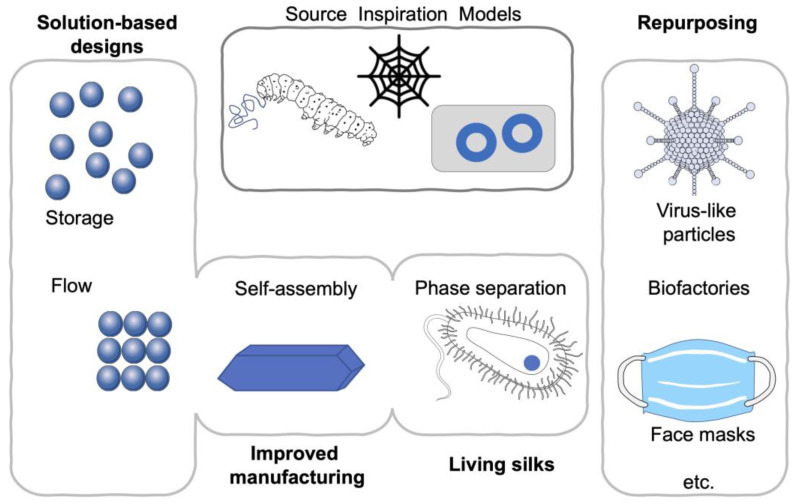
Summary of emerging silk material trends. Repurposing [23,25,27,28], solution-based designs [24], phase separation [30,31] and flow [22,26,29] impacting manufacturing and creating living materials.

## Data Availability

No new data were created or analyzed in this study. Data sharing is not applicable to this article.

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
