# Peer review of "Emerging Silk Material Trends: Repurposing, Phase Separation and Solution-Based Designs"

_materials, 2021, doi:10.3390/ma14051160_

Round 1

Reviewer 1 Report

Overall, I find the manuscript really interesting. I like the logic frame and the fact that not only pure science is taken into consideration. All the provided details and the selection of key papers prove the well-established knowledge of Dr. Seib, who confirms to be a key scientist in the silk community.

I enjoyed reading the paper and I was truly impressed by its fluency in reading.

Despite this, I really have minor concerns about it.

  • Being a review, I would really like to see a nice figure (key one perhaps) that provides an aesthetic point of view of the situation and related to some key devices.
  • Reference 57, Kaplan must be written with the capital letter.
  • Since spider silk is barely cited, what are the author thoughts about it in comparison to silkworm silk? In fact, from the natural material point of view, it is clear that there are major differences. But what about the recombinant, processed and liquid materials? Does the author have any thoughts about this? Does the author think that some general statements about both silks’ types can be done? If so, does the author think that is worth it to cite here such similarities?

Author Response

I would like to thank the reviewer for the kind and very supportive words. Thank you so much for the careful review of this manuscript. Your valuable suggestions have helped me to improve this work. Specifically, I have:

(1) created an original figure.

(2) corrected this oversight.

(3) Thank you for your suggestions in relation to the wider silk space. I have implemented this, please see lines 106 to 120 (tracked document).

Reviewer 2 Report

The manuscript is well written and clear. It describes the history of silk use, silk formation and silk material. My one criticism would be a complete lack of any figures which would make the manuscript more approachable to a wider audience. Line 23 cites silk has arisen 15 time from an 1997 paper however Sutherland et a. 2010l perform an extensive analysis and state 23 times.

Author Response

I would like to thank the reviewer for the very supportive words and helpful comments. I have created an original figure, now use the up-to-date Sutherland publication and have correct the number too.

Reviewer 3 Report

The Covid 19 section at the end seems forced, and I would remove it.

There are other liquid and phase separation papers of silk you should reference:

Tseng P, Directed assembly of bio-inspired hierarchical materials with controlled nanofibrillar architectures. Nature Nanotechnology 12, 474-480, 2017

You discuss the phase-separation and solution based design of silk, I would change the title of the manuscript to reflect the description. 

Author Response

I would like to thank the reviewer for the very helpful comments. I have updated the title of the manuscript and include the 2017 Tseng study (lines 323 to 329 tracked document).